# Clinical Influence of Ethanol Infusion in the Vein of Marshall on Left Atrial Appendage Occlusion: Results of Feasibility and Safety during Implantation and at 60-Day Follow-Up

**DOI:** 10.3390/jcm12051960

**Published:** 2023-03-01

**Authors:** Yibo Ma, Miaoyang Hu, Lanyan Guo, Jian Xu, Jie Li, Qun Yan, Huani Pang, Jinshui Wang, Ping Yang, Fu Yi

**Affiliations:** 1Department of Cardiology, Xijing Hospital, Air Force Medical University, Xi’an 710032, China; 2Department of Cardiology, The First Hospital of Hanbin District, Ankang 725000, China; 3Department of Cardiology, Baoji People’s Hospital, Baoji 721006, China

**Keywords:** atrial fibrillation, vein of Marshall, left atrial appendage occlusion, peri-device leak, cardiac computed tomography angiography

## Abstract

Background: Ethanol infusion in the vein of Marshall (EI-VOM) has the advantages of reducing the burden of atrial fibrillation (AF), decreasing AF recurrence, and facilitating left pulmonary vein isolation and mitral isthmus bidirectional conduction block. Moreover, it can lead to prominent edema of the coumadin ridge and atrial infarction. Whether these lesions will affect the efficacy and safety of left atrial appendage occlusion (LAAO) has not yet been reported. Objectives: To explore the clinical outcome of EI-VOM on LAAO during implantation and after 60 days of follow-up. Methods: A total of 100 consecutive patients who underwent radiofrequency catheter ablation combined with LAAO were enrolled in this study. Patients who also underwent EI-VOM at the same period of LAAO were assigned to group 1 (*n* = 26), and those who did not undergo EI-VOM were assigned to group 2 (*n* = 74). The feasibility outcomes included intra-procedural LAAO parameters and follow-up LAAO results involving device-related thrombus, a peri-device leak (PDL), and adequate occlusion (defined as a PDL ≤ 5 mm). Safety outcomes were defined as the composites of severe adverse events and cardiac function. Outpatient follow-up was performed 60 days post-procedure. Results: Intra-procedural LAAO parameters, including the rate of device reselection, rate of device redeployment, rate of intra-procedural PDLs, and total LAAO time, were comparable between groups. Furthermore, intra-procedural adequate occlusion was achieved in all patients. After a median of 68 days, 94 (94.0%) patients received their first radiographic examination. Device-related thrombus was not detected in the follow-up populations. The incidence of follow-up PDLs was similar between the two groups (28.0% vs. 33.3%, *p* = 0.803). The incidence of adequate occlusion was comparable between groups (96.0% vs. 98.6%, *p* = 0.463). In group 1, none of the patients experienced severe adverse events. Ethanol infusion significantly reduced the right atrial diameter. Conclusions: The present study showed that undergoing an EI-VOM procedure did not impact the operation or effectiveness of LAAO. Combining EI-VOM with LAAO was safe and effective.

## 1. Introduction

Atrial fibrillation (AF), one of the most common arrhythmias, can lead to a variety of disabling and fatal complications, particularly stroke and systemic embolism [1]. Left atrial appendage occlusion (LAAO), a nonpharmacological treatment for thromboprophylaxis, has been proven to be non-inferior to warfarin or novel oral anticoagulants in high-risk AF populations [2,3,4]. Because both LAAO and radiofrequency catheter ablation (RFCA) are percutaneous interventional procedures and need to be performed in a catheterization laboratory, many operators combine both procedures in a single procedure to not only achieve long-term stroke prevention without lifeline anticoagulation but also to acquire effective rhythm control. Previous studies have shown that combining LAAO and RFCA can reduce procedure costs without affecting the long-term efficacy of the individual procedures, and their combination does not increase the incidence of procedure-related complications [5,6].

RFCA alone may be insufficient to achieve ideal rhythm control in persistent AF patients because electrical pulmonary vein isolation (PVI) is difficult to achieve, and the benefits of additional linear ablation or complex fractionated electrogram ablation are unclear [1,7,8]. Moreover, ethanol infusion in the vein of Marshall (EI-VOM) is a novel treatment for persistent AF. It has the advantage of eliminating AF triggers, facilitating a PVI and bidirectional peri-mitral block, achieving local denervation, and cutting the pathological conduction branches [9]. In addition, EI-VOM in combination with RFCA can lead to better rhythm control in patients with persistent AF without excessive complications compared to RFCA alone [10]. In a single-arm study, it was shown that EI-VOM combined with RFCA significantly improved left atrial function [11]. Thus, EI-VOM should be a potential therapeutic measure.

It should be noted that EI-VOM can generate prominent edema of the coumadin ridge and lead to atrial infarction. Whether these lesions will affect the efficacy and safety of LAAO has not yet been reported. Several concerns need to be addressed: (1) whether EI-VOM affects implantation of the closure device; (2) whether EI-VOM leads to a greater leak; and (3) whether patients will experience more clinical adverse events. To address these concerns, this study was developed to explore the clinical influence of EI-VOM on LAAO during implantation and after 60 days of follow-up.

## 2. Materials and Methods

### 2.1. Study Populations

This study was a single-center, retrospective cohort study. One hundred consecutive patients who underwent a one-step procedure (RFCA combined with LAAO) were included from February 2020 to August 2022 at the First Affiliated Hospital of Air Force Medical University. Patients were enrolled if they met the following criteria: (1) age between 18 and 85 years; (2) finished at least 60 days of radiographic follow-up; and (3) intracardiac echocardiography-guided LAAO. Exclusion criteria were as follows (1) valvular AF, including but not limited to moderate or severe mitral stenosis; (2) alcohol allergy; (3) a left atrial diameter of greater than 65 mm. This study adhered to the principles of the Declaration of Helsinki and was approved by the institutional ethics committee of the First Affiliated Hospital of Air Force Medical University. Each patient signed an informed written consent form before the procedure.

### 2.2. Procedure

#### 2.2.1. Ethanol Infusion in the Vein of Marshall

EI-VOM was performed as previously described [9,10,11]. Firstly, a 6F guide catheter was advanced into the coronary sinus to obtain a venogram and to identify the opening of the vein of Marshall (VOM) (Figure 1A). If found, an angiography wire was advanced through the 6F catheter and into the VOM, and subsequently, an OTW balloon (2 mm * 8 mm) was advanced over the wire and positioned in the ostium of the VOM. A second coronary sinus venogram was performed after the balloon was inflated to reveal the anatomy of the VOM (Figure 1B). Next, the balloon was positioned on the distal part of the VOM. After the inflation of the balloon, the first ethanol injection was performed. Finally, the balloon was deflated and returned to the proximal end of the VOM, during which inflation and ethanol injection were repeated approximately 4 times. Contrast infiltration indicated procedure success (Figure 1C). After EI-VOM, RFCA and electrophysiological studies were performed.

#### 2.2.2. Left Atrial Appendage Occlusion

Intracardiac echocardiography (ICE)-guided LAAO was performed as previously described [12]. In brief, the radiofrequency catheter was replaced with a device delivery sheath. An ICE probe was advanced into the left atrium along with the device delivery sheath through the same atrial septal puncture hole. Left atrial appendage (LAA) morphology was first acquired to confirm that the device that was selected before the procedure was appropriate. Then, the closure device was implanted under the guidance of ICE. During occlusion, prominent edema of the coumadin ridge was seen (Figure 1D). Both a Watchman device (Boston Scientific) and a LACbes device (PushMed, Shanghai, China) were used as alternatives. If occlusion was appropriate, the closure device was released. At the end of the LAAO procedure, hemostasis was ensured by manual compression.

### 2.3. Clinical Outcomes and Follow-Up

In this study, clinical outcomes included feasibility and safety outcomes. The feasibility outcomes included intra-procedural LAAO parameters and follow-up LAAO results. Intra-procedural LAAO parameters included the successful implantation of the first selected device, device redeployment, intra-procedural peri-device leak (PDL), intra-procedural adequate occlusion (defined as a PDL ≤ 5 mm), and total LAAO procedure time (defined as the duration from the first angiography to device release). Follow-up LAAO results included device-related thrombus (DRT), PDL, and adequate occlusion. The safety outcome in this study was the composite of severe adverse events (SAEs) within 30 days [5,13,14]. SAEs included death, ischemic or hemorrhagic stroke, systemic embolism, transient ischemia attack, air or device embolism, pericardial tamponade requiring drainage, and major bleeding requiring transfusion. Changes in cardiac function before and after EI-VOM served as an additional safety outcome. Patients were treated with novel oral anticoagulants post-procedure, and additional single antiplatelet therapy was prescribed to patients who recently underwent coronary revascularization. Reporting on clinical outcomes occurred during periprocedural care until 60 days of follow-up post-procedure.

### 2.4. Computed Tomography Assessment

Cardiac computed tomography angiography (CCTA) was preferred in detecting DRT or PDL, the details of which have been previously described [15]. A second-generation, dual-source Computed Tomography scanner (Somatom Definition Flash, Siemens Healthcare) was used to show the closure device, LAA, and other adjacent structures. First, a non-contrast Computed Tomography scan was performed. Then, a volume of 50 mL contrast medium was injected, and a region of interest was selected. Once ideal scan acquisition timing was achieved, an arterial phase scan was executed. A venous phase scan (delayed imaging) was performed 30–60 s after the start of the arterial phase scan. The image data was stored on a DVD-R and evaluated by both imaging experts and cardiologists. Finally, the results were integrated. In case the patient refused to receive a CCTA examination, TEE was used as an alternative.

DRT was detected on the arterial and venous phase images and distinguished from the metal artifact on non-contrast images. LAA patency was observed on the venous phase images and defined as LAA density ≥ 100 Hounsfield units or ≥ 150% of that measured at the same site for the arterial phase [13]. If LAA patency was present, PDL was defined as the passage of contrast medium along the margins of the closure device (Figure 2).

### 2.5. Statistical Analysis

Statistical analysis was performed using R 4.2.0 (Robert Gentleman & Ross Ihaka, Auckland, New Zealand) and SPSS Statistics 26.0 (IBM SPSS Inc., Chicago, USA). Normality distribution was examined by the Shapiro-Wilk test. Continuous variables with a normal distribution are described as the mean ± standard deviation and were compared using a 1-way analysis of variance or paired t-test. Non-normally distributed continuous variables are expressed as the median with interquartile range (IQR) and were compared using the Mann-Whitney U test. Categorical variables are described as counts (percentages) and were compared using the Chi-square test or Fisher’s exact probability test according to positive rates. A 2-sided *p*-value ≤ 0.05 was considered statistically significant.

## 3. Results

### 3.1. Baseline Characteristics

A total of 100 consecutive patients were enrolled in this study, with 26 patients in group 1 (underwent EI-VOM) and 74 patients in group 2 (did not undergo EI-VOM), respectively. Baseline characteristics were consistent between the two groups. The mean age was 64.1 ± 9.0 years, and the majority were male. The medians of the CHA_2_DS_2_-VASc score and HASBLED score were 3.0 (IQR: 2.0, 4.0) and 2.0 (IQR: 1.0, 3.0), respectively. In both groups, most patients had persistent AF. Twenty-three patients (23.0%) had a history of cardiac embolism, and 15 patients (15.0%) had a history of non-cardiac stroke. The details of the baseline characteristics are listed in Table 1.

### 3.2. Intra-Procedural Left Atrial Appendage Occlusion Parameters

Periprocedural characteristics are summarized in Table 2. The majority morphology of the LAA was cauliflower, and the orifice diameter and depth of LAA were comparable between groups. Most patients in both groups chose a Watchman device.

All intra-procedural LAAO parameters were comparable (Table 2). One patient (3.8%) in group 1 and five patients (6.8%) in group 2 underwent at least two different device implantation attempts (*p* = 1.000). Seven patients (26.9%) in group 1 and 33 patients (44.6%) in group 2 had their devices redeployed before final release (*p* = 0.162). Adequate intra-procedural device deployment was achieved in all patients, and ICE and fluoroscopy confirmed that five (19.2%) and 13 (17.6%) patients had PDLs < 5 mm in groups 1 and 2, respectively (*p* = 1.000). Three patients (11.5%) in group 1 and seven patients (7.7%) in group 2 had <3 mm intra-procedural PDLs (*p* = 0.717); two patients (9.5%) in group 1 and six patients (8.1%) in group 2 had 3–5 mm intra-procedural PDLs (*p* = 1.000). The procedure time spent on LAAO in groups 1 and 2 was 24.0 (IQR: 14.0, 34.0) min and 20.0 (16.0, 28.0) min, respectively (*p* = 0.553).

### 3.3. Follow-Up Left Atrial Appendage Occlusion Results

After a median of 68 (IQR: 59, 89) days, 25 patients (96.2%) in group 1 and 69 patients (93.2%) in group 2 received their first radiographic examinations (Table 3). A total of 66 patients (70.2%) received a CCTA examination. More patients in group 1 chose CCTA as their follow-up examination (88.5% vs. 58.1%, *p* = 0.005). Follow-up results were comparable between the two groups: DRT or suspicious DRT was not found in the follow-up cohort; seven (28.0%) and 23 (33.3%) patients had PDLs in groups 1 and 2, respectively (*p* = 0.803); two patients (8.0%) in group 1 and 11 patients (15.9%) in group 2 had < 3 mm PDLs (*p* = 0.502); four patients (16.0%) in group 1 and 11 patients (15.9%) in group 2 had 3–5 mm PDLs (*p* = 1.000); and 24 patients (96.0%) in group 1 and 68 patients (98.6%) in group 2 achieved adequate occlusion (*p* = 0.463). The LAA patency in patients who received CCTA follow-up was also evaluated (Figure 3). A total of 31 patients (52.2%) had patency LAA, with 11 patients (51.3%) in group 1 and 21 patients (52.3%) in group 2 (Fisher exact test, *p* = 1.000).

### 3.4. Safety ourcomes

In group 1, none of the patients experienced SAEs. In group 2, one patient (1.4%) experienced major bleeding. The patient improved after the transfusion.

Cardiac function was evaluated in 25 follow-up patients before and after EI-VOM, and the results are presented in Figure 4. After 2 months of follow-up, the post-procedural right atrial diameter was significantly reduced (42.5 ± 5.4 vs. 40.0 ± 5.9, *p* = 0.02), while the left atrial diameter (46.3 ± 6.1 vs. 45.0 ± 4.7, *p* = 0.23) and left ventricular ejection fraction (56.7 ± 5.9 vs. 56.2 ± 5.0, *p* = 0.64) did not significantly different between pre-procedure and post-procedure.

## 4. Discussion

In this study, the clinical influence of EI-VOM on LAAO was reported during implantation and after 60 days of follow-up. The following was found: (1) EI-VOM did not make the LAAO procedure more difficult because all intra-procedural LAAO parameters were comparable between the two groups; (2) group-to-group differences in follow-up LAAO results were comparable; and (3) in group 1, none of the patients experienced SAEs. Taken together, these results show that combining EI-VOM with LAAO is feasible and safe.

### 4.1. Intra-Procedural Feasibility

Electrophysiologists face the same challenges, whether performing the one-step procedure or a combination of EI-VOM, RFCA, and LAAO. For the one-step procedure, the main problem operators face is edema of the coumadin ridge [6]. Phillips et al. reported their 5-year single-center experience with the one-step procedure. Although coumadin ridge edema was common, adequate occlusion was achieved in all patients, thereby suggesting that edema did not affect the LAAO operation [16]. In fact, many studies have reported that the one-step procedure did not decrease adequate occlusion rates when compared to LAAO alone [6]. It is worth noting that the coumadin ridge is the corresponding endocardial structure of the ligament of Marshall, so EI-VOM combined with RFCA could lead to more severe edema than RFCA alone (Figure 2D). However, in our study, we demonstrated that EI-VOM did not affect the implantation of the closure device.

In the present study, it was shown that the rate of follow-up PDLs was increased compared to intra-procedural PDLs in both group 1 and group 2. This might be related to the dissipation of the coumadin ridge edema. This indicates that edema of the coumadin ridge has a substantial influence on PDL. Unfortunately, the degree of edema was not measured. So far, no accurate measure of coumadin ridge edema has been published [17]. Attempts at further comparisons of edema pre- and post-procedure or between groups were abandoned in our study. When performing the three-dimensional reconstruction of the left atrial pre-EI-VOM, the ICE probe was placed in the right atrial area. While executing the LAAO operation, the ICE probe was deployed in the left atrium. This resulted in the lack of comparability of measurements pre- and post-procedure. In addition to PVI, ablation targets in our study varied because of differences in the atrial substrate. This resulted in the lack of comparability of measurements between patients. Therefore, the results for intra-procedural feasibility are relatively crude. The development of a standard measure program will be required to further explore what degree of edema will occur and its effects.

### 4.2. Follow-Up Feasibility

In recent years, several studies have reported the value of CCTA in LAAO procedure follow-up. Zhao et al. reported their single-center experience in detecting incomplete occlusion by CCTA after the one-step procedure [18]. In the SWISS-APERO trial, both TEE and CCTA were used simultaneously as follow-up examinations after LAAO [13]. The results of both studies showed that the sensitivity of CCTA in detecting a leak was better than that of TEE. Considering the above, in this study, CCTA was chosen as the primary follow-up examination. CCTA could be detected as a very small PDL, as shown in Figure 2B. Our study revealed that the EI-VOM procedure did not affect the follow-up results for LAAO: DRT or suspicious DRT was not observed in our study; the occlusion rate was adequate, and PDLs were comparable between groups. When considering the degree of PDL, no statistical significance was observed between groups. In addition, adequate occlusion was comparable between groups, and so was patency LAA. As previously mentioned, edema of the coumadin ridge has a substantial influence on PDL. Furthermore, atrial infarction, which was caused by ethanol injection, may also affect the incidence and size of the PDL. The VENUS trial showed that EI-VOM could damage 4.9 ± 3.2 cm^2^ of atrial tissue on average [10], which included part of the LAA tissue. LAA infarction might increase the incidence and rate of PDL; it may also prevent the endothelialization of the closure device and even cause acute and delayed cardiac tamponade. One of our primary concerns was whether tissue loss affects the efficacy of the procedure. We are currently providing a possibility that the damage caused by EI-VOM will not increase the incidence and degree of PDL, as well as the incidence of patency LAA.

The PDL in our study was comparable with that presented in previous reports. Zhao et al. reported that the 6-month PDL in the CCTA examination was 39.3% (33/84). The SWISS-APERO trial showed that the total incidence of peri-device leakages (PDL and mixed leak) in the Amulet group and Watchman group were 22.9% (24/105) and 34.0% (34/100), respectively. These results also prove that EI-VOM will not increase the risk of PDL.

### 4.3. Safety Outcome

As mentioned earlier, safety considerations for atrial infarction were another critical concern. The present study showed that none of the patients experienced pericardial effusion or tamponade. Thus, our results do not indicate an increased risk of pericardial effusion caused by atrial infarction.

In previous studies, the safety of the one-step procedure has been widely reported, and the RECORD real-world study showed that combining catheter ablation, whether using radiofrequency or a cryoballoon, with LAAO did not increase the risk of SAEs [5,6]. The safety of EI-VOM combined with RFCA has also been reported in previous studies. Valderrábano et al. showed that the blood alcohol level after EI-VOM was undetectable at regular doses [19]. The VENUS trial and a real-world study showed that combining EI-VOM with RFCA did not increase the risk of clinical adverse events [10,15]. Chen et al. shared their single-center experience with a one-step procedure. More than 1000 patients were enrolled, and the safety and efficacy were supported. Patients who underwent EI-VOM combined with LAAO were also included. However, these patients were not analyzed separately [20]. Our results showed that combining EI-VOM with LAAO did not increase the risk of clinical adverse events. In group 1, none of the patients experienced SAEs. Recently, the concept of a same-day discharge after LAAO was proposed, and its effectiveness and safety were preliminarily verified [21,22]. These findings provide a clear direction for our further clinical practice.

In addition, the short-term changes in cardiac function pre- and post-procedure were evaluated. According to the echocardiographic findings, ethanol infusion significantly reduced the right atrial diameter. Derval et al. reported their single-center experience in EI-VOM combined with RFCA; the atrial function also served as an additional feasibility endpoint. Significant improvements in transmitral A-wave and ejection fraction were observed at day 1 and month 12 post-ablation [11]. These improvements may be attributed to better rhythm control. It has been widely verified that EI-VOM combined with RFCA significantly reduced the risk of arrhythmia recurrence compared to RFCA alone [10,23,24]. Further follow-up is needed to determine the long-term improvements in the left atrial diameter and left ventricular ejection fraction.

### 4.4. Limitations

There were several limitations to this study. The sample size of the study was small, and the ratio of the intervention group (group 1) to the control group (group 2) was 1:3, which may limit the power to detect potential differences. We abandoned EI-VOM for those patients who experienced VOM dissection intra-procedure in order to not increase the risk of pericardial effusion [14]. The use of two completely different types of closure devices may introduce confounding bias.

## 5. Conclusions

Combining ethanol infusion in the vein of Marshall and left atrial appendage occlusion is feasible and safe. The intra-procedure parameters and follow-up results were comparable with those of the one-step procedure both in our cohort as well as in previous reports. This combination did not increase the risk of clinical adverse events. Furthermore, ethanol infusion significantly reduced the right atrial diameter. Large-scale and long-term prospective clinical follow-up is warranted to further complement our findings.

## Figures and Tables

**Figure 1 jcm-12-01960-f001:**
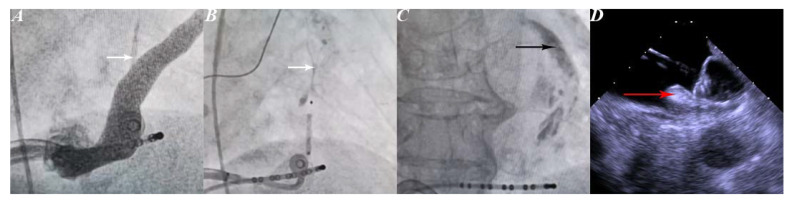
Steps of the procedure. Panel (**A**) VOM (white arrow) in Coronary sinus venogram. Panel (**B**) Vein of Marshall venogram was performed after the balloon was inflated to reveal the anatomy of the VOM (white arrow). Panel (**C**) Contrast infiltration (black arrow). Panel (**D**) Watchman closure implantation. Prominent edema of the coumadin ridge (red arrow) in the echocardiographic image. VOM, vein of Marshall.

**Figure 2 jcm-12-01960-f002:**
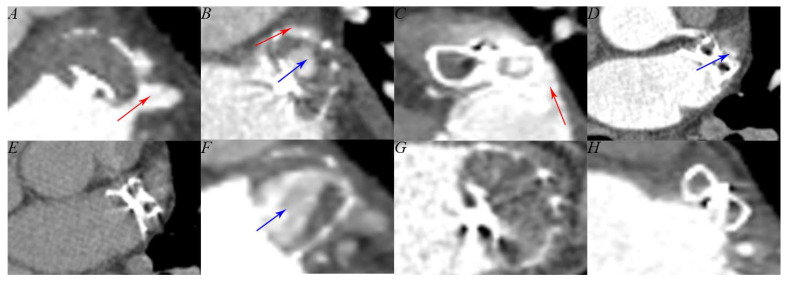
The patterns of left atrial appendage patency. Panel (**A**) Peri-device leak (red arrow) only in Watchman device. Panel (**B**) Peri-device leak (red arrow) with an intra-device leak (blue arrow) in Watchman device. Panel (**C**) Peri-device leak (red arrow) only in LACbes device. Panel (**D**) Intra-device leak (blue arrow) only in LACbes device. Panel (**E**) No contrast image of panel (**D**). Panel (**F**) Intra-device leak only in Watchman device (blue arrow). Panel (**G**) Non-patency left atrial appendage in Watchman device. Panel (**H**) Non-patency left atrial appendage in LACbes device.

**Figure 3 jcm-12-01960-f003:**
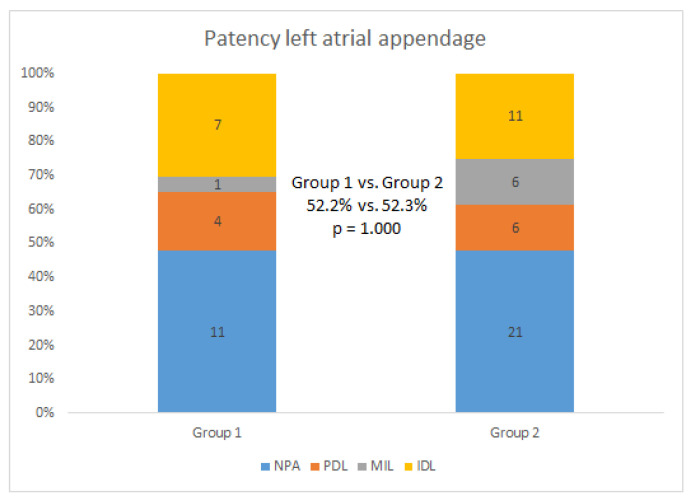
Follow-up results in patients who received cardiac computed tomography angiography. NPA = no patency left atrial appendage; PDL = peri-device leak; MIL = mixed leak; IDL = intra-device leak.

**Figure 4 jcm-12-01960-f004:**
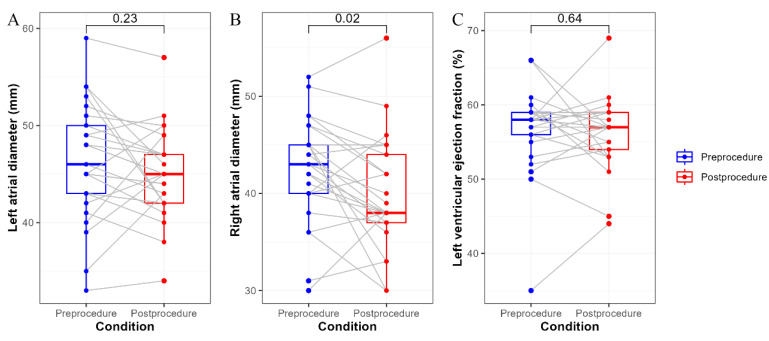
Cardiac function pre- and post-ethanol infusion. Panel (**A**) Changes in the left atrial diameter; Panel (**B**) Changes in the right atrial diameter; Panel (**C**) Changes in the left ventricular ejection fraction. Variables were compared using the paired *t*-test.

**Table 1 jcm-12-01960-t001:** Patient characteristics at baseline.

	Group 1 (*n* = 26)	Group 2 (*n* = 74)	*p*-Value
**Demographics**			
Male sex	14 (53.8)	51 (68.9)	0.232
Age, years	65.4 ± 10.4	63.8 ± 8.6	0.553
BMI, kg/m^2^	25.0 ± 2.8	25.5 ± 3.1	0.447
**AF overview**			
CHA_2_DS_2_-VASc score	3.0 (2.0, 4.0)	3.0 (2.0, 4.0)	0.674
HASBLED score	2.0 (1.0, 3.0)	2.0 (1.0, 3.0)	0.794
AF pattern			0.353
Paroxysmal	8 (30.8)	32 (43.2)	–
Non-paroxysmal	18 (69.2)	42 (56.8)	–
AF duration, months	24.0 (5.0, 48.0)	12.0 (4.0, 36.0)	0.544
**Comorbidities**			
Heart failure	8 (30.8)	20 (27.0)	0.801
Hypertension	13 (50.0)	42 (56.8)	0.648
Type Ⅱ diabetes	5 (19.2)	15 (20.3)	1.000
Previous stroke/TIA/SE	11 (42.3)	27 (36.5)	0.643
Coronary heart disease	6 (23.1)	25 (33.8)	0.460
Bleeding tendency	6 (23.1)	16 (21.6)	1.000
**Echocardiographic index**			
Left atrial diameter, mm	46.2 ± 6.0	44.7 ± 5.6	0.247
Left ventricular ejection fraction, %	56.5 ± 5.9	55.9 ± 5.9	0.651

BMI = body mass index; AF = atrial fibrillation; TIA = transient ischemic attack; SE = systemic embolism.

**Table 2 jcm-12-01960-t002:** Periprocedural characteristics and feasibility.

	Group 1 (*n* = 26)	Group 2 (*n* = 74)	*p*-Value
**Periprocedural characteristics**			
LAA morphology			0.346
Cauliflower	20 (76.9)	53 (71.6)	–
Chicken wing	2 (7.7)	10 (13.5)	–
Reversed chicken wing	3 (11.5)	2 (2.7)	–
Windsock	1 (3.8)	7 (9.5)	–
Cactus	0 (0.0)	2 (2.7)	–
LAA ostia diameter, mm	22.9 ± 4.0	23.0 ± 3.8	0.942
LAA depth, mm	23.6 ± 4.7	24.1 ± 3.1	0.571 ^1^
Device			0.073
Watchman	18 (69.2)	64 (86.5)	–
LACbes	8 (30.8)	10 (13.5)	–
**Periprocedural feasibility**			
LAAO time, min	24.0 (14.0, 34.0)	20.0 (16.0, 28.0)	0.553
LAAO success	26 (100.0)	74 (100.0)	1.000
Device reselection	1 (3.8)	5 (6.8)	1.000
Redeployment	7 (26.9)	33 (44.6)	0.162
Incidence of PDL	5 (19.2)	13 (17.6)	1.000
<3 mm jet size	3 (11.5)	7 (9.5)	0.717
3–5 mm jet size	2 (7.7)	6 (8.1)	1.000
Mean size, mm	2.4 ± 1.1	2.6 ± 0.7	0.539

^1^ Patients who underwent LACbes implantation were excluded when evaluating the mean depth. LAA = left atrial appendage; LAAO = left atrial appendage occlusion; PDL = peri-device leak.

**Table 3 jcm-12-01960-t003:** Follow-up feasibility.

	Group 1 (*n* = 25)	Group 2 (*n* = 69)	*p*-Value
Time to review, days	69.0 (60.0, 87.0)	67.0 (57.0, 88.0)	0.827
Radiographic examination			0.005
CCTA follow-up	23 (88.5)	43 (58.1)	–
TEE follow-up	2 (11.5)	26 (35.1) ^1^	–
DRT or suspicious DRT	0 (0.0)	0 (0.0)	1.000
Adequate occlusion	24 (96.0)	68 (98.6)	0.463
PDL	7 (28.0)	23 (33.3)	0.803
<3 mm jet size	2 (8.0)	11 (15.9)	0.502
3–5 mm jet size	4 (16.0)	11 (15.9)	1.000

^1^ One patient received both TEE and CCTA. CCTA = cardiac computed tomography angiography; TEE = transesophageal echocardiography; DRT = device-related thrombus; PDL = peri-device leak.

## Data Availability

The datasets used and/or analyzed during the current study can be available from the corresponding author upon reasonable request.

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
