# Peer review of "Clinical Influence of Ethanol Infusion in the Vein of Marshall on Left Atrial Appendage Occlusion: Results of Feasibility and Safety during Implantation and at 60-Day Follow-Up"

_jcm, 2023, doi:10.3390/jcm12051960_

Round 1

Reviewer 1 Report

In this paper, Ma et al. present the effects of ethanol infusion in the vein of Marshall in patients with atrial fibrillation undergoing Left Atrial Appendage Occlusion. The authors adopted a retrospective cohort study design enrolling 100 patients, 26 in the intervention group (ethanol infusion together with LAAO) and 74 in the control group (LAAO alone). The authors reported no difference in outcomes between the two groups. While the study is well designed and assesses an important topic, there are certain issues that should be addressed:

1) The authors have adopted a 1:3 ratio of intervention to control group (26 over 74). This severely limits the power of the study and may not allow the detection of differences that may potentially exist. An adequate justification of this study design is needed.

2) While the goal of the study is to assess safety, it would be important to assess whether ethanol infusion has any effects on atrial function. Please report the before and after intervention atrial function outcomes for the study population.

3) A few grammar and spelling issues were identified throughout the article. Please perform a grammar check prior to resubmiting.

Author Response

Dear professor:

I am very grateful to your comments for the manuscript. According with your advice, we amended the relevant part in manuscript. Some of your questions were answered below.

Response to Reviewer 1 Comments

Point 1: The authors have adopted a 1:3 ratio of intervention to control group (26 over 74). This severely limits the power of the study and may not allow the detection of differences that may potentially exist. An adequate justification of this study design is needed.

Response 1: Thank you very much for your comment. We know that this problem can seriously affect the stability of the results. However, the pandemic of COVID-19 infections is causing substantial disruptions to our daily work and life. We have reported all patients who underwent the combination of Marshall vein ethanol infusion, radiofrequency ablation and left atrial appendage occlusion. Originally, we want to report the study as a case series. But given the lack of controls and the fact that patients in the cohort were primarily received CT follow-up, we chose all patients who received radiofrequency ablation combined with left atrial appendage occlusion during the same period as the control group. We have mentioned this as limitation.

Point 2: While the goal of the study is to assess safety, it would be important to assess whether ethanol infusion has any effects on atrial function. Please report the before and after intervention atrial function outcomes for the study population.

Response 2: Thank you very much for your valuable suggestions. We have evaluated and reported the left atrial diameter, right atrial diameter, and left ventricular ejection fraction pre- and post-procedure. Besides, we use the discussion session for the results explanation.

Point 3: A few grammar and spelling issues were identified throughout the article. Please perform a grammar check prior to resubmitting.

Response 3: Thank you very much for your reminding. Before resubmitting, we checked the manuscript for spelling and grammar as best we could.

Best regards,

Yibo Ma

Reviewer 2 Report

Congratulations on taking on this interesting topic. I don't have any major objections, only in my opinion there is a lack of information about the indications for closing the left atrial appendage, especially since these were not mostly high-risk bleeding patients. Moreover, considering the endpoints, it is interesting to know whether the type of occluder used was related to their occurrence. In addition, with regard to the endpoints, it is important whether the patients were taking NOACs all the time until follow-up. 

Author Response

Dear professor:

I am very grateful to your comments for the manuscript. According with your advice, we amended the relevant part in manuscript. Some of your questions were answered below.

Response to Reviewer 2 Comments

Point 1: Congratulations on taking on this interesting topic. I don't have any major objections, only in my opinion there is a lack of information about the indications for closing the left atrial appendage, especially since these were not mostly high-risk bleeding patients. Moreover, considering the endpoints, it is interesting to know whether the type of occluder used was related to their occurrence. In addition, with regard to the endpoints, it is important whether the patients were taking NOACs all the time until follow-up.

Response 1: Thank you very much for your comment. (i) The indication of LAAC was according to the local guidelines, which can be summarized as patients unwilling to receive long-term OAC, intolerance to anticoagulation, or at high risk of embolism. (ii) The type of closure device used might not related to the endpoints. Take follow-up peri-device leakage for example, the incidence of leakage was comparable between WM patients and LACbes patients (p = 0.255). However, not many patients received LACbes implantation. This may limit the power to detect any potential differences. (iii) Patients were treated with NOACs post-procedure, and additional single antiplatelet therapy was prescribed to patients who recently underwent coronary revascularization. Excepted for patients lost to follow-up, every patient had good medication compliance.

Best regards,

Yibo Ma

Round 2

Reviewer 1 Report

The authors have addressed sufficiently all the comments made by the reviewer. Pending some final grammatical and vocabulary corrections (eg section 3.4, line 7), I recommend accepting this manuscript for publication.

Author Response

Dear professor:

I am very grateful to your comments for the manuscript. According with your advice, we amended the relevant part in manuscript. Some of your questions were answered below.

Response to Reviewer 1 Comments

Point 1: The authors have addressed sufficiently all the comments made by the reviewer. Pending some final grammatical and vocabulary corrections (eg section 3.4, line 7), I recommend accepting this manuscript for publication.

Response 1: Thank you very much for your reminding. Before resubmitting, we finally checked the manuscript for spelling and grammar as best we could.

Best regards,

Yibo Ma
